# HyperRAG: Hierarchy-Aware Retrieval-Augmented Generation with Hyperbolic Embeddings for Ontology-Based Entity Linking

## Abstract

Extracting structured knowledge from unstructured text is a fundamental challenge in machine learning, particularly when the target concepts are organized within complex hierarchical ontologies. We present HyperRAG, a novel framework that integrates Large Language Models (LLMs) with Retrieval-Augmented Generation (RAG) and hierarchical reranking using hyperbolic embeddings. Our approach is designed to improve entity linking and retrieval in settings where the label space exhibits rich hierarchical relationships. In addition, we introduce a hierarchy-aware evaluation framework that leverages ontology structure to provide a more nuanced assessment of model performance, moving beyond conventional exact-match metrics. Through comprehensive experiments on both benchmark and real-world datasets, including a newly curated and challenging set of clinical notes for phenotype extraction in precision medicine, we demonstrate that HyperRAG substantially improves ranking accuracy and recall, especially for implicit or nuanced entity mentions. While our primary application is in the biomedical domain, the proposed framework is broadly applicable and generalizable to hierarchical entity linking and retrieval tasks in other domains. All code, models, and datasets are released to support reproducibility.

## 1 Introduction

Extracting structured knowledge from unstructured text is a central challenge in machine learning, particularly when the target concepts are organized within complex hierarchical ontologies (Fig. 1). This challenge is especially pronounced in precision medicine, where linking clinical observations to standardized phenotype concepts is critical for diagnosis, treatment planning, and biomedical research (Son et al., 2018; Yuan et al., 2022; Mao et al., 2025). While recent advances in natural language processing have improved the extraction of explicitly stated phenotypes, existing systems often struggle to identify implicit or context-dependent mentions.

A key limitation of current approaches (Feng et al., 2023; Luo et al., 2021; Arbabi et al., 2019) is their reliance on flat embedding spaces, which are not well-suited for modeling the hierarchical relationships inherent in ontologies such as the Human Phenotype Ontology (HPO) (Robinson et al., 2008). In contrast, hyperbolic geometry naturally models hierarchical (tree-like) structures (Gromov, 1987) and has been underexplored in this context. Furthermore, retrieval-based systems are often constrained by their reliance on exact matches or shallow semantic representations. We also argue that existing evaluation metrics (Groza et al., 2024) present further limitations: in practice, clinicians may interpret phenotype mentions differently, as no individual possesses exhaustive knowledge of HPO or uses it in a uniform manner. This observation extends to all domains where annotations rely on complex ontologies, as annotators may vary in their familiarity and interpretation of hierarchical structures. Consequently, a single reference can yield multiple, equally valid annotations. This underscores the importance of considering hierarchical relationships, such as treating a parent term of a target mention as correct, albeit less specific.

In this paper, our contributions are four-fold:

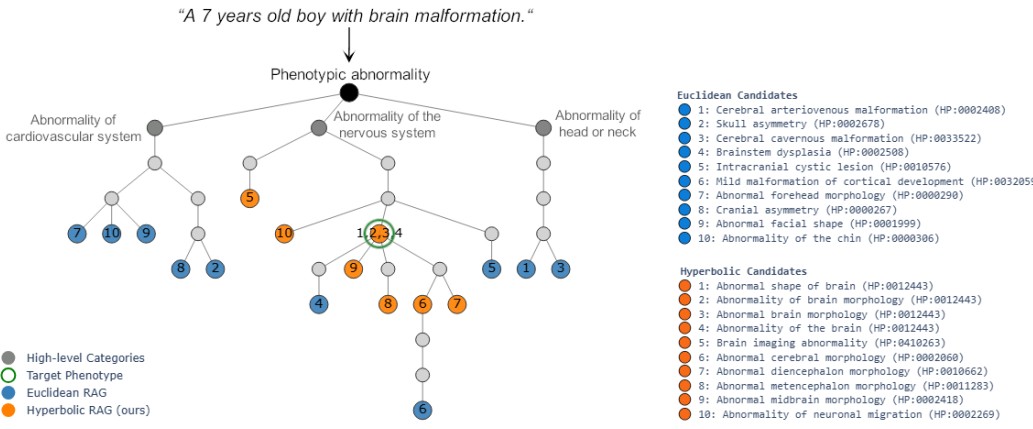

Figure 1: Illustration of ontology-based entity linking for a given clinical sentence. A subset of the Human Phenotype Ontology (HPO) hierarchy is shown, with the target phenotype highlighted in green. The top-10 candidate phenotypes retrieved by Euclidean RAG (blue) and Hyperbolic RAG (orange) are overlaid. Notably, all hyperbolic candidates are located within the correct target branch, and their average distance (number of hops) to the target is substantially lower than that of Euclidean candidates. This demonstrates that hyperbolic embeddings yield candidates that are more semantically and hierarchically aligned with the target compared to standard Euclidean approaches.

    i. We propose a workflow for hierarchical entity linking that integrates Large Language Models (LLMs) with Retrieval-Augmented Generation (RAG) and hierarchical reranking using hyperbolic embeddings trained on target ontologies.

    ii. We introduce a hierarchy-aware evaluation framework that leverages the target ontology structure to provide a more nuanced and robust assessment of extraction systems.

    iii. We demonstrate the effectiveness of our approach through comprehensive experiments on both benchmark and challenging real-world datasets, showing substantial improvements, particularly in scenarios with implicit entity mentions.

    iv. We release all generated training and evaluation datasets, as well as the trained models, to support reproducibility and further research in ontology-based entity linking.

Our empirical analysis confirms that hyperbolic embeddings effectively capture hierarchical relationships, supporting their use in downstream entity linking tasks. While our primary application is in the biomedical domain, the proposed framework is broadly applicable to hierarchical entity linking and retrieval tasks in other domains, offering a robust and generalizable solution for extracting structured knowledge from text.

## 2 RELATED WORK

The introduction of ontologies such as the HPO has provided a structured framework for organizing phenotypic information and has become the primary target for entity linking in this domain. Early work utilized rule-based heuristics (Aronson & Lang, 2010; Jonquet et al., 2009; Deisseroth et al., 2019), while more recent studies have adopted transformer-based architectures to extract phenotype mentions directly from text (Luo et al., 2021; Feng et al., 2023; Yang et al., 2024). Although improvements have been effective with such approaches, they remain complex and often struggle when phenotype references are implicit (Baddour et al., 2024). In particular, full LLM approaches are prone to hallucination issues in mapping phenotype labels to HPO ids (Labbé et al., 2023). Emerging paradigms such as RAG (Lewis et al., 2020) offer a promising avenue for addressing some of these challenges by efficiently narrowing the candidate space and mitigating hallucinations. However, RAG has not yet been widely adopted in phenotype extraction pipelines, and its performance in this context remains underexplored.

While ontologies facilitate annotation and retrieval, their hierarchical complexity poses significant challenges for NLP systems. Nickel & Kiela (2017) highlighted the limitations of flat embedding spaces in adequately representing such hierarchical structures. Related works (Sala et al., 2018; Sinha et al., 2024; Tifrea et al., 2018) proposed to train hyperbolic embeddings that provide a compelling alternative, as hyperbolic spaces are well-suited for modeling hierarchical relationships, allowing embeddings to more accurately reflect the subsumption structure inherent in ontologies. These challenges and solutions are not unique to the biomedical domain, but are broadly relevant to any field where structured ontologies underpin annotation and retrieval tasks.

The motivation behind our proposed workflow stems from recognizing significant limitations in current phenotype extraction systems. While classical RAG approaches (Guo et al., 2024) are effective at retrieving candidates based on general semantic similarity, they fall short in capturing the hierarchical relationships and intricate dependencies inherent in ontologies such as HPO (Huang et al., 2025; Gao et al., 2023). This limitation becomes even more pronounced when dealing with implicit phenotypes not explicitly stated in clinical text, where leveraging ontological relationships can be crucial for accurate identification and resolution (Peng et al., 2024).

In the following section, we describe our approach, which addresses these limitations by leveraging hyperbolic embeddings and hierarchical reranking.

## 3 METHODOLOGY

The HyperRAG workflow is illustrated in Figure 2: Given clinical reports, the process consists of four main steps: span detection using an LLM, candidate retrieval with RAG, reranking of candidates, and evaluation with both standard and ontology-aware metrics.

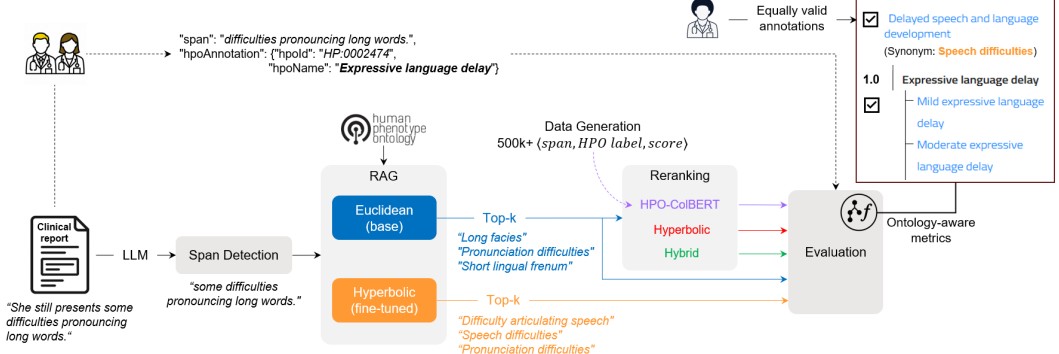

Figure 2: General Workflow. From a clinical report, a LLM first detects spans related to phenotypes. RAG with Euclidean and Hyperbolic embeddings retrieves *Top-k* candidate terms, followed by several reranking strategies. Evaluation uses both clinician-provided ground truth annotations and the HPO ontology structure to account for hierarchical relationships and semantic similarity, recognizing that multiple valid references may exist.

### 3.1 SPAN DETECTION AND CANDIDATES RETRIEVAL

*Span Identification*

We use a pretrained LLM to identify phenotype spans, effectively capturing implicit mentions that may be overlooked by traditional methods. Notably, Baddour et al. (2024) demonstrated that employing an LLM as a span detector outperforms the biomedical Stanza pipeline (Zhang et al., 2021).

*Retrieval-Augmented Generation (RAG)*

We compute dense span embeddings using either a base or HPO-fine-tuned hyperbolic model. *Top-k* phenotype candidates are retrieved from the HPO ontology based on *cosine similarity* (euclidean model) or normalized *hyperbolic distance* (hyperbolic model).

## 3.2 RERANKING

For each span, the *Top-k* candidates retrieved by Euclidean-RAG are reranked using two families of methods: late-interaction and hyperbolic-based.

*Late-interaction reranking*

As a strong classical baseline, we fine-tuned a late-interaction model (Santhanam et al., 2021) for reranking. While cross-encoders are effective, they are computationally demanding and less adaptable to incorporating soft signals like distance-based scores (Jha et al., 2024). Late-interaction models strike a balance by retaining token-level embeddings and applying a late matching function, thus preserving fine-grained information often lost in bi-encoders. This makes them particularly suitable for our reranking task, which involves short text spans and specific target labels.

*Hyperbolic-based reranking*

- **Full hyperbolic reranking:** Both the input span and the *Top-k* candidates from the Euclidean RAG are embedded in hyperbolic space. Candidates are then reordered based on their normalized hyperbolic distances to the input span.

- **Hybrid reranking:** This approach combines the cosine similarity between Euclidean embeddings and the hyperbolic distance between hyperbolic embeddings using a weighted sum. Cosine similarity emphasizes semantic closeness, while hyperbolic distance prioritizes candidates with closer hierarchical relationships to the input span. Given an input $span$ and a set of candidates $\{C_i\}_{i=1}^{k}$, the hybrid score of a candidate $C_i$ is computed as follows:

$$S_{\text{hybrid}}(C_i, span) = \gamma \cdot S_{\cos} - (1 - \gamma) \cdot \hat{d}_{\mathbb{H}} \tag{1}$$

where $S_{\cos}$ is the cosine similarity between $C_i$ and $span$, $\hat{d}_{\mathbb{H}}$ is their normalized hyperbolic distance, and $\gamma$ is the weighting parameter between the two metrics (see Appendix C for ablation).

Full implementation details (including model architectures, training hyperparameters, normalization strategies, synonym mapping, and retrieval settings) are provided in Appendix A and B to facilitate reproducibility.

Although our experiments focus on biomedical ontologies, the HyperRAG workflow is ontology-agnostic and can be directly applied to any domain with a structured hierarchical ontology, such as product taxonomies or legal codes.

## 4 DATA

### 4.1 ONTOLOGIES

The Human Phenotype Ontology (HPO) (Robinson et al., 2008) serves as the foundation for our hierarchical embeddings, comprising over 19,000 hierarchically organized phenotypic terms and an extensive set of synonyms.

Additionally, we leverage the SNOMED ontology (El-Sappagh et al., 2018) indirectly through a pretrained hyperbolic model (fine-tuned from the same base model). It allows us to assess the relative benefits of utilizing this broad, general-purpose medical ontology in comparison to a highly specialized ontology (HPO) for phenotype extraction tasks.

### 4.2 TRAINING DATA

*Hyperbolic training*: Hyperbolic embeddings were trained on HPO triplets extracted from the ontology, with synonym augmentation and filtering to ensure balanced training. We used Hierarchy Transformers (He et al., 2024) framework to effectively capture hierarchical relationships.

Training parameters are provided in Appendix B.

*Late-interaction training*: We fine-tuned ColBERTv2 (Santhanam et al., 2021) on triplets ⟨*span, HPO label, score*⟩, using a dataset generated by prompting ChatGPT-4o-mini to create diverse clinical sentences and corresponding phenotype spans for each HPO term. Sentence quality was assessed by clinicians, and further validated using an LLM-as-a-judge approach. After heuristic filtering, we obtained 91,760 high-quality spans, which were used to construct positive and negative pairs, resulting in 510,371 training triplets.

Full training parameters and data construction details for both hyperbolic and late-interaction models are provided in Section 9 and Appendix B. Prompts for LLM-as-a-judge are provided in Appendix G.

### 4.3 EVALUATION DATASET

We evaluate our workflow using two datasets. **ID-68** (Anazi et al., 2017), A widely used benchmark for phenotype extraction. **CHU-50**, an internal dataset containing 50 manually generated synthetical clinical notes from a partner hospital containing a total of 971 phenotype annotations, approximately 30% of which are implicit and much more challenging. Results are compared to PhenoBERT (Feng et al., 2023), the most advanced open-source state-of-the-art solution available.

## 5 EVALUATION

We first evaluated the hyperbolic model independently, prior to conducting the main phenotype extraction experiments.

### 5.1 HYPERBOLIC INNER EVALUATION

To assess the consistency of the hyperbolic model, we compared its normalized distance metrics with those of the baseline Euclidean model. Specifically, we examined one-hop and multi-hop distances to evaluate the model's ability to capture hierarchical relationships, as well as distances between synonyms and negative pairs to determine whether semantic consistency is preserved.

Additionally, we introduce a *hierarchical representation power* plot to visualize the model's capacity to encode hierarchy while maintaining semantic coherence. This radar chart displays the average distances for one-hop, multi-hop, and synonym pairs, alongside the inverse average distance for negative pairs. This visualization enables us to assess whether the embedding space has been structured as intended.

### 5.2 PHENOTYPES LINKING EVALUATION

In practice, generating a comprehensive list of phenotypes for each patient is crucial for accurate diagnosis, making recall-based metrics (*recall@k* and *miss_rate@k*) the primary focus. While Top-1 precision is reported for comparison with existing methods, it can be biased by clinician habits and is less informative at higher ranks. To further assess ranking quality, we include Mean Reciprocal Rank (MRR) and Normalized Discounted Cumulative Gain (NDCG).

However, these traditional metrics are limited when based solely on exact matches, which is the prevailing evaluation paradigm in current solutions. In practice, a parent term of the target phenotype often conveys relevant information, even if it is less specific, and predictions involving descendants or related terms should not be considered entirely incorrect.

To address this limitation, we introduce a novel hierarchical evaluation framework that leverages the structure of HPO to weight candidate scores according to their proximity to the ground truth. These *relationship scores* are computed based on the specific type of relationship between the candidate $\mathcal{C}$ and the target phenotype $\mathcal{T}$.

*Direct relationships*

$$w_{direct}(\mathcal{C}, \mathcal{T}) = \begin{cases} \frac{\alpha}{p \times (1+|d|)}, & p > 0 \\ 1, & p = 0 \end{cases} \quad (4)$$

where $\alpha$ is a constant factor, $p$ is the number of ancestors/descendants between $\mathcal{C}$ and $\mathcal{T}$, and $d$ is the distance between $\mathcal{C}$ and $\mathcal{T}$.

*Indirect relationships*

$$w_{indirect} = \frac{\beta}{c \times (1 + d_l)} \quad (5)$$

where $\beta$ is a constant factor, $c$ is the number of immediate descendants of the most specific common ancestor between $\mathcal{C}$ and $\mathcal{T}$, and $d_l$ is the distance between $\mathcal{C}$ and the farthest HPO leaf.

See Appendix C for $\alpha$ and $\beta$ settings.

By combining absolute distances with the cardinality of surrounding phenotypes, these functions effectively characterize the strength of relationships between HPO terms, balancing both proximity and semantic relevance. Throughout this paper, the term *weighted* refers to evaluation metrics that incorporate these hierarchical weightings.

In addition, we introduce specific metrics to assess how well the models respect the ontology's structure: the average number of hops between each candidate and the target phenotype; the average branch coverage, defined as the proportion of candidates within the same branch as the target; and the distribution of relationship types by position, measuring the proportions of exact matches, ancestors, descendants, cousins, or candidates with no direct path to the target. We also report the proportion of close candidates, defined as those with a *relationship score* above a specified threshold.

## 6 RESULTS

### 6.1 HYPERBOLIC CONSISTENCY

Figure 3. presents the distributions of one-hop (a) and multi-hop (b) distances for both the Euclidean model and the fine-tuned hyperbolic model. The distributions for the hyperbolic model are notably narrower and exhibit lower means, particularly for multi-hop distances, indicating a more faithful representation of the ontology's hierarchical structure.

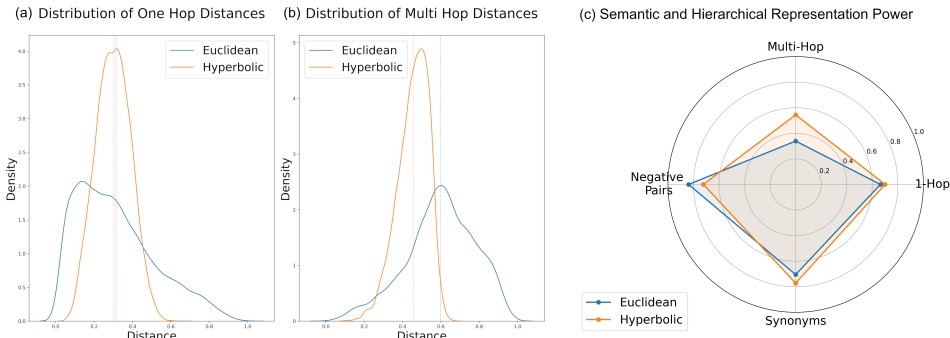

Figure 3: Euclidean (blue) vs Hyperbolic (orange) distance distributions between one-hop (a) and multi-hop (b) phenotypes in the Human Phenotypes Ontology (HPO), computed on the test set (10%). Vertical lines represent respective means. (c) represents normalized similarity (or inverse for negative pairs) between one-hop, multi-hop, and synonyms in Euclidean and Hyperbolic spaces.

Furthermore, the resulting hyperbolic model preserves the semantic structure of the base model, as illustrated in Figure 3 (c). Although the average distance between negative pairs is slightly reduced, these pairs remain well separated from positive examples. Notably, synonyms within the

HPO are now positioned closer together, and multi-hop phenotypes are significantly closer than in the Euclidean embedding space, reflecting improved hierarchical modeling. In contrast, one-hop phenotypes are only marginally closer, which is expected given the typically strong semantic similarity between such terms (e.g.: *Iris coloboma* is semantically closer to its one-hop parent *Coloboma* than the 2-hops *Abnormal eye morphology*).

## 6.2 PHENOTYPES LINKING

For the hybrid approach, we conducted evaluations with several values of $\gamma$ (reported in Appendix C). Since the performance differences were marginal, we selected $\gamma = 0.5$ for its simplicity and interpretability, recognizing that this value may not be optimal but provides a balanced influence of both models. The results reported in this section are based on this default value.

**Retrieval**

As shown in Figure 4, the hyperbolic RAG model underperforms compared to other methods on both datasets, with recall further decreasing when hyperbolic reranking is applied to Euclidean RAG candidates.

For ID-68 (top plots), hybrid reranking improves recall from k=5 onwards, and late-interaction reranking becomes effective from k=15, though it does not surpass the hybrid approach. The Euclidean model remains a robust baseline for recall, achieving a precision at k=1 of 0.857. Weighted recall follows similar trends, with ontology-aware metrics particularly benefiting hyperbolic models and narrowing performance gaps. Both Euclidean and hybrid reranking outperform previous SOTA recall from k=3 onwards and set new SOTA at k=1 in the weighted setting (+9), with gains up to +18 at k=15. Hybrid reranking also achieves the lowest miss rate, reducing misses by 17 at (k=15).

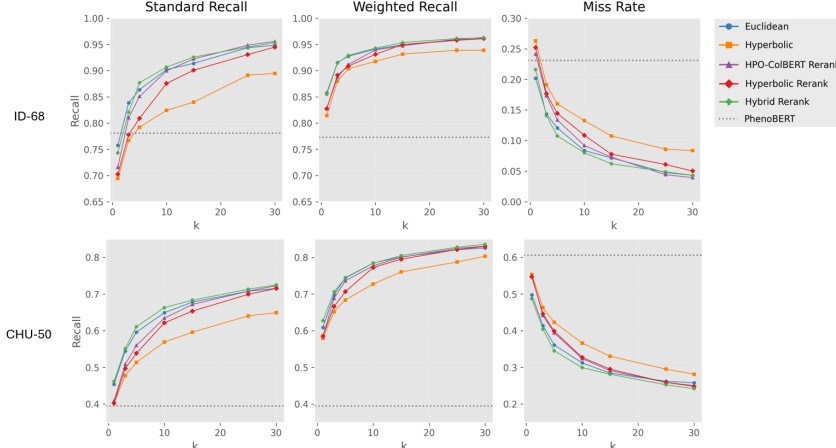

Figure 4: Recall and miss rate on the ID-68 (top) and CHU-50 (bottom) datasets. Standard (exact match) and weighted recall (considering HPO hierarchy) are shown for raw RAG outputs (Euclidean, Hyperbolic) and reranked results (HPO-ColBERT, Hyperbolic, Hybrid) across *Top-k* candidates. Miss rate indicates the proportion of ground truth phenotypes not retrieved.

On CHU-50 (bottom plots), similar trends are observed, with all models outperforming state-of-the-art PhenoBERT across all three metrics from k=1, achieving a +23 increase in recall and an 18-point reduction in miss rate. This is consistent with previous findings (Baddour et al., 2024), as PhenoBERT is less effective at capturing implicit phenotype references. Overall, the logarithmic shape of the recall and miss rate curves across both datasets indicates that all models rank correct candidates highly.

**Ranking**

Table 1 presents weighted MRR and NDCG results for ID-68 and CHU-50. On ID-68, the Euclidean and Hybrid Rerank models achieve the highest MRR, indicating top-ranked correct phenotypes. The Hyperbolic model performs slightly lower, but the gap narrows with hierarchy-aware metrics,

highlighting its strength in capturing ontological relationships. NDCG scores are also high across all models, with Euclidean and Hybrid Rerank exceeding 0.94 at k=1. As k increases, both metrics decrease slightly, but Hybrid Rerank consistently maintains strong performance.

Table 1: Weighted Metrics by Model and *k* for ID-68 and CHU-50

| | ID-68 | | | | | CHU-50 | | | | |
|---|---|---|---|---|---|---|---|---|---|---|
| | **Weighted MRR** | | | | | **Weighted MRR** | | | | |
| Model | $k=1$ | $k=3$ | $k=5$ | $k=10$ | $k=15$ | $k=1$ | $k=3$ | $k=5$ | $k=10$ | $k=15$ |
| Euclidean | **0.857** | **0.882** | **0.884** | **0.885** | **0.885** | 0.609 | 0.643 | 0.651 | 0.655 | 0.656 |
| Hyperbolic | 0.814 | 0.841 | 0.845 | 0.847 | 0.848 | 0.579 | 0.607 | 0.612 | 0.616 | 0.617 |
| HPO-ColBERT Rerank | 0.828 | 0.851 | 0.855 | 0.858 | 0.858 | 0.585 | 0.624 | 0.633 | 0.636 | 0.638 |
| Hyperbolic Rerank | 0.828 | 0.853 | 0.857 | 0.859 | 0.860 | 0.585 | 0.616 | 0.623 | 0.629 | 0.631 |
| Hybrid Rerank | **0.855** | **0.881** | **0.883** | **0.885** | **0.885** | **0.627** | **0.656** | **0.662** | **0.666** | **0.667** |
| | **Weighted NDCG** | | | | | **Weighted NDCG** | | | | |
| Model | $k=1$ | $k=3$ | $k=5$ | $k=10$ | $k=15$ | $k=1$ | $k=3$ | $k=5$ | $k=10$ | $k=15$ |
| Euclidean | **0.949** | **0.936** | **0.923** | **0.901** | 0.883 | 0.842 | **0.838** | 0.816 | **0.791** | 0.777 |
| Hyperbolic | 0.932 | 0.922 | 0.902 | 0.877 | 0.861 | 0.844 | 0.824 | 0.800 | 0.770 | 0.747 |
| HPO-ColBERT Rerank | 0.931 | 0.928 | 0.912 | 0.886 | 0.870 | 0.844 | 0.821 | 0.805 | 0.783 | 0.768 |
| Hyperbolic Rerank | 0.944 | 0.927 | 0.912 | 0.884 | 0.870 | 0.850 | 0.832 | 0.805 | 0.772 | 0.763 |
| Hybrid Rerank | 0.943 | **0.936** | **0.924** | **0.904** | **0.891** | **0.861** | **0.841** | **0.824** | **0.794** | **0.784** |

On the more challenging CHU-50 dataset, which contains a higher proportion of implicit phenotype mentions, all models exhibited lower MRR and NDCG scores compared to ID-68. However, the Hybrid Rerank model outperforms others for all k values.

These results indicate that the hybrid approach, which combines semantic similarity from Euclidean embeddings with hierarchical proximity from hyperbolic embeddings, is particularly effective in ranking the most relevant phenotypes at the top, even in complex, real-world clinical text.

Overall, the consistently strong MRR and NDCG scores for the Hybrid Rerank model confirm that combining semantic and hierarchical signals yields superior candidate ranking. Hierarchy-aware weighted metrics further demonstrate the value of hyperbolic embeddings in capturing nuanced ontological relationships, especially when exact matches are unavailable but related terms remain clinically relevant.

**Ontology-based Metrics**

Analyzing the number of hops between candidates and the target phenotype, as well as branch coverage (Figure 5), offers further insight into model performance. While hyperbolic-based models may not always outperform Euclidean models at top ranks, they show greater robustness as the candidate list grows, maintaining lower average hops and higher branch coverage.

This observation is further supported by a detailed analysis of ontological relationships (Appendix D). Figure 6 shows a high percentage of exact matches at Top-1, confirming the RAG pipeline's effectiveness. However, deeper analysis reveals important distinctions between modeling approaches. Hyperbolic models (both raw output and reranking) exhibit significantly higher proportions of ancestor and

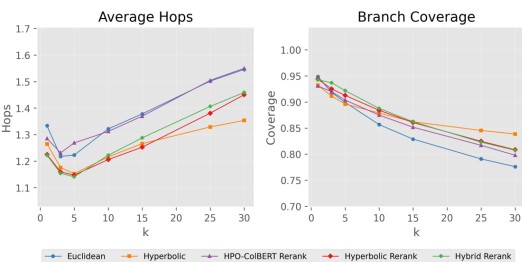

Figure 5: Ontology-based metrics on the ID-68 dataset across *Top-k* candidates. Average number of hops between the candidates and the target phenotypes (left), and proportion of candidates within the ontology branch of the target phenotypes (right).

cousin relationships, while showing fewer descendant relationships compared to Euclidean or HPO-ColBERT models. This pattern strongly suggests that hyperbolic approaches better capture the hierarchical structure of the HPO ontology in both vertical and horizontal dimensions. The tendency to "move upward" in the hierarchy toward more general terms rather than "downward" toward more specific ones aligns with theoretical expectations of hyperbolic geometry, where distances increase exponentially with depth in the hierarchy.

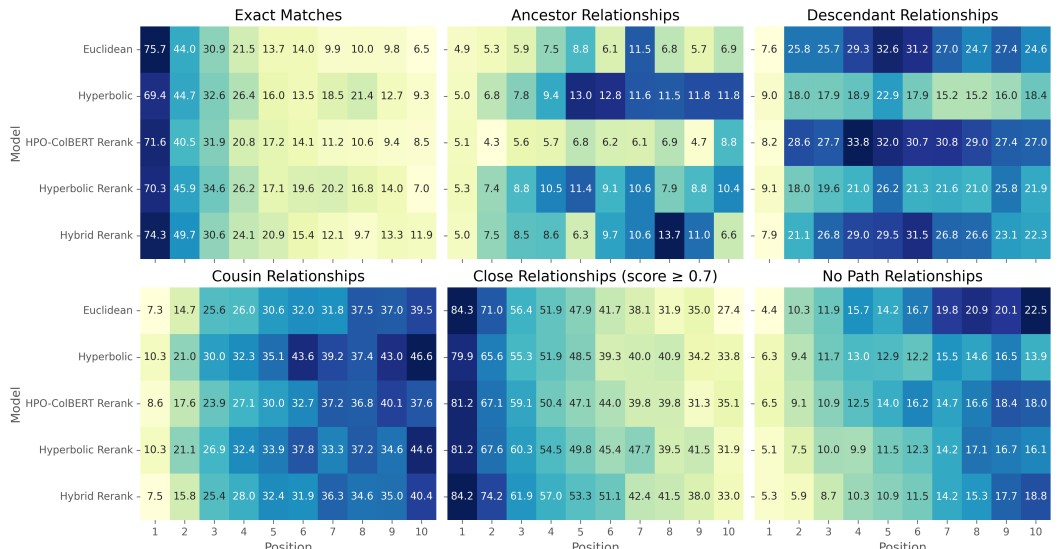

Figure 6: Distribution of relationship types by position and model for Top-10 candidates (k=10) on ID-68 dataset. The heatmaps show percentages for six relationship categories (Appendix D) between predicted and ground truth phenotypes. For closeness relationships, only those with scores above 0.7 (Equations equation 4 and equation 5) are considered. Rows correspond to different models, and columns represent candidate positions ranked from 1 to 10. Higher values indicate stronger presence of the relationship type at the given rank and model.

Notably, hyperbolic models maintain semantic relevance at higher ranks, preserving close relationships and yielding fewer unrelated candidates as k increases. This semantic consistency at higher ranks has important implications for clinical applications, as it reduces the risk of missing relevant phenotypes (false negatives) when examining a broader set of candidates.

The hybrid reranking approach combines the strengths of both geometries, achieving strong exact matching at top positions and semantic coherence at higher ranks. This balanced performance confirms the value of integrating both approaches for optimal phenotype retrieval in clinical settings. Similar trends are observed on the CHU-50 dataset (Figure 15 in Appendix F).

## 7 CONCLUSION

In summary, this work introduces HyperRAG, a novel pipeline that synergistically combines LLM-based span detection, retrieval-augmented generation, and hierarchical reranking using hyperbolic embeddings for phenotype linking from clinical text. Comprehensive experiments on benchmark and real-world datasets demonstrate substantial improvements in recall, miss rate, and ranking quality, especially with hierarchy-aware metrics that better reflect clinical relevance. The hybrid reranking strategy, integrating both semantic and ontological signals, consistently delivers state-of-the-art performance, especially in scenarios with implicit phenotype mentions. The proposed evaluation framework and publicly released datasets further advance the field by enabling more nuanced and clinically meaningful assessment of phenotype extraction systems. Future work should focus on enhancing the semantic modeling of implicit mentions, expanding to additional ontologies, and optimizing the computational efficiency of the pipeline for broader adoption. Overall, given its reliance only on the existence of a hierarchical ontology and unstructured text, HyperRAG is readily applicable to a wide range of domains beyond biomedicine, such as e-commerce, law, and scientific literature.

## 8 ETHICS STATEMENT

The CHU-50 dataset introduced in this work consists of clinical notes with phenotype annotations. All data were manually generated by clinicians based on real clinical reports, but the content is fully original and anonymized. No personal data are present in the dataset, and care was taken to ensure that no combination of symptoms could indirectly identify any individual. This approach ensures compliance with ethical standards and protects patient privacy.

Additionally, the proposed workflow relies in part on probabilistic outputs, such as those produced by LLMs. These outputs should be interpreted with caution. If used in a medical context, all results must be reviewed and validated by qualified healthcare professionals before any clinical decision is made.

## 9 REPRODUCIBILITY STATEMENT

Code, model and datasets are available at: https://

In addition, we provide hereafter the detailed description of dataset construction steps.

*Hyperbolic Training Data*

Hierarchical relationships are extracted from the HPO OWL file using DeepOnto and the ELK reasoner. Following the methodology of He et al. (2024), we generated a dataset of triplets ⟨*child, parent, label*⟩, where the label is a binary indicator of a positive or negative example, and triplets ⟨*child, parent, negative*⟩, where the negative term is not a parent of the child. We used random negative sampling strategy in this implementation. Given that most HPO phenotypes are associated with multiple synonyms, we augmented the dataset by including all possible synonym combinations within each triplet. This augmentation enhances the robustness of the resulting embeddings to varied term formulations. To prevent excessive class imbalance, we applied a filtering strategy, limiting each synonym to a maximum of five occurrences.

*Late-interaction Training Data*

We fine-tuned the ColBERTv2 model (Santhanam et al., 2021) on triplets of the form ⟨*span, HPO label, score*⟩, where the score represents a similarity measure. To construct a comprehensive training dataset, we first used ChatGPT-4o-mini to generate 10 clinical report sentences for each HPO term in the ontology. To ensure diversity and representativeness, we specified requirements for each batch of 10 sentences (e.g., at least two sentences should be implicit, up to two should include measurements, etc.). For this iteration, we excluded cases where a sentence refers to multiple phenotypes. For each generated sentence, we further prompted ChatGPT-4o-mini to extract the most precise span capturing the clinical observation of the target phenotype. This process resulted in the *HPO_sentences_spans* dataset, comprising over 200,000 clinical sentences and corresponding spans, covering the entire set of HPO terms. All prompts are provided in Appendix G for reproducibility.

| Name | Definition |
|---|---|
| WHITELIST_WORDS | Set of medically relevant or specific terms; if any are present in a span, the span is always kept. |
| BLACKLIST_PATTERNS | Set of phrases indicating non-informative or undesirable content; if present (and not overridden by whitelist), the span is always filtered out. |
| GENERIC_WORDS | Set of common, non-specific words; used to penalize spans that are mostly generic. |
| VAGUE_PATTERNS | Set of vague or non-specific phrases; presence reduces the span's score. |

Table 2: Heuristic lists used for span filtering.

After qualitative analysis of generated spans, some of them appear to be uninformative (e.g.: for the sentence "*The bladder capacity measured at 150 mL, which is below the normal range.*", the

output span is "*below the normal range*" which lacks context to be relevant). We conducted a deep qualitative analysis of the spans with our clinicians in order to define filtering rules. Table 2 define the heuristics elements used in the process. The scoring procedure and the span filtering are detailed in pseudo-code in Algorithm 1 and Algorithm 2 respectively.

---

**Algorithm 1** Compute Span Score

---

**Require:** span (string)
1: **if** span contains any **BLACKLIST_PATTERN then**
2:    **if** span contains any **WHITELIST_WORD then**
3:       **return** 1.0
4:    **else**
5:       **return** 0.0
6:    **end if**
7: **end if**
8: **if** span contains any **WHITELIST_WORD then**
9:    **return** 1.0
10: **end if**
11: **if** number of words in span $\geq 3$ **then**
12:    score $\leftarrow 1.0$
13: **else**
14:    score $\leftarrow 0.3$
15: **end if**
16: generic_ratio $\leftarrow$ (number of **GENERIC_WORDS** in span) / (total words)
17: **if** generic_ratio $> 0.6$ **then**
18:    score $\leftarrow$ score $-0.5$
19: **end if**
20: **if** span contains any **VAGUE_PATTERN then**
21:    score $\leftarrow$ score $-0.3$
22: **end if**
23: **return** max(score, 0.0)

---

**Algorithm 2** Filter Spans

---

**Require:** spans (list of strings)
1: **for** each span in spans **do**
2:    score $\leftarrow$ **Compute Span Score**(span)
3:    **if** score $\geq 0.2$ **then**
4:       keep span
5:    **else**
6:       filter out span
7:    **end if**
8: **end for**

---

As a result, this lower-quality spans filtering yields 91,760 unique high-quality spans (with 2,167 unique spans filtered out).

Finally, we leveraged the trained hyperbolic model for scoring: positive *(span, HPO label)* pairs from the generated dataset were assigned a score of 1, while negative pairs were created by pairing spans with other phenotypes and assigning scores based on the normalized hyperbolic distance to the target phenotype. Both hard negatives (phenotypes within the same branch, up to three hops away) and easy negatives (phenotypes outside the target branch) were included. The final training set consists of 510,371 triplets ⟨*span, HPO label, score*⟩.

ACKNOWLEDGMENTS

We acknowledge the use of LLM (*ChatGPT-4.1-mini*) for minor writing assistance, specifically grammar and wording suggestions. The authors remain fully responsible for all scientific content.

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

## A    Implementation Details

*Span detection*: for consistency with Baddour et al. (2024) and comprehensive coverage, we utilized ChatGPT-3.5 model (OpenAI, 2023). Corresponding prompt is described in Appendix G.

*RAG*: we used *all-MiniLM-L12-v2* (Wang, 2020) as the embeddings base model, from which the hyperbolic model was fine-tuned. We set *k=30* to substantially reduce the candidate space while still allowing for meaningful reranking improvements.

*Candidate retrieval*: FAISS (Douze et al., 2024) is employed as the vector store and *Top-k* retriever for the Euclidean model. In contrast, the hyperbolic model utilizes a dedicated vector index and retrieval mechanism.

To ensure consistency in distance measurements across experiments, we normalize the hyperbolic distances in the Poincaré ball using a global normalization strategy equation 2:

$$\hat{d}_{\mathbb{H}}(u,v) = \frac{d_{\mathbb{H}}(u,v)}{\max_{p,q \in \text{HPO}} d_{\mathbb{H}}(p,q)} \tag{2}$$

where $d_{\mathbb{H}}(u,v)$ is the hyperbolic distance between terms $u$ and $v$ in the hyperbolic space $\mathbb{H}$, $\hat{d}_{\mathbb{H}}$ is the normalized hyperbolic distance, and $\max_{p,q \in \text{HPO}} d_{\mathbb{H}}(p,q)$ is the maximum hyperbolic distance between two terms in the HPO ontology.

Synonyms in the RAG output are mapped to their original HPO terms using a precomputed synonym-to-ID mapping. This ensures consistency in distance calculations throughout the workflow.

## B    Training Settings

All model training was conducted on a single RTX A3000 GPU, both to accommodate budget constraints and to reduce energy consumption for environmental considerations.

Table 3 indicates the training settings for the hyperbolic model training.

| Parameter | Value |
|---|---|
| Number of training epochs | 20 |
| Train batch size | 32 |
| Eval batch size | 64 |
| Learning rate | 1e-5 |
| Clustering loss weight | 1.0 |
| Clustering loss margin | 5.0 |
| Centripetal loss weight | 1.0 |
| Centripetal loss margin | 0.5 |
| Gradient accumulation steps | 8 |

Table 3: Hyperbolic Training Hyperparameters.

The training settings for the ColbertV2 fine-tuning on HPO is presented in Table 4.

| Parameter | Value |
|---|---|
| Train batch size | 8 |
| Learning rate | 1e-5 |
| Number of training epochs | 2 |
| Max query length | 32 |
| Max document length | 128 |
| Triplet loss margin | 0.3 |
| Gradient accumulation steps | 2 |

Table 4: ColBERTv2 Training Hyperparameters.

## C PARAMETERS SETTINGS

For the hierarchy-aware metrics, the parameters $\alpha$ (Equation 4) and $\beta$ (Equation 5) were empirically set to 1.6 and 1.0, respectively, through random assessment by clinicians, who reviewed diverse samples to ensure clinical relevance.

Regarding $\gamma$ parameter (Equation 1), we conducted additional ablation studies varying the ratio between Euclidean and hyperbolic components in the hybrid model to assess their respective contributions for the ID-68 and CHU-50 dataset. As shown in Figures 7 and 8, our findings indicate that except for very small values of $\gamma$ (which consistently underperform), recall performance shows no clear trends across the different $\gamma$ settings. Among them, $\gamma = 0.5$ and $\gamma = 0.7$ produce the best overall results.

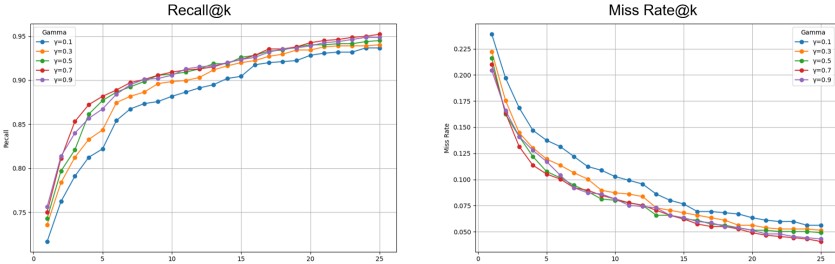

Figure 7: Recall@k and Miss Rate@k for the ID-68 dataset for different values of the weighting parameter $\gamma$, which balances Euclidean and hyperbolic distances in the hybrid reranking score.

Interestingly, for the CHU-50 dataset (characterized by a high proportion of implicit references) the hybrid model with $\gamma = 0.5$ performs identically to, and sometimes better than, $\gamma = 0.7$. This suggests that, in such cases, the hyperbolic component is more effectively leveraged, likely due to its ability to capture hierarchical or indirect relationships.

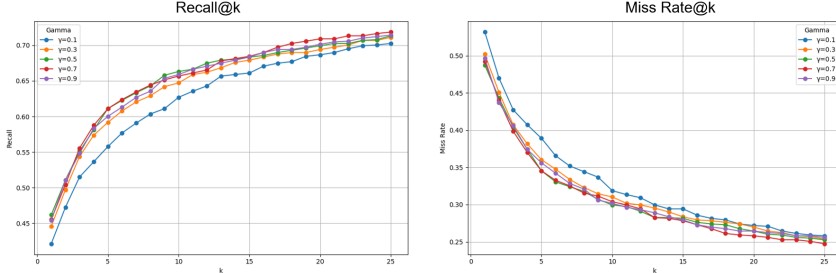

Figure 8: Recall@k and Miss Rate@k for the CHU-50 dataset for different values of the weighting parameter $\gamma$, which balances Euclidean and hyperbolic distances in the hybrid reranking score.

Given these marginal differences, and to maintain simplicity and interpretability, we fix $\gamma = 0.5$ in the main experiments presented in the paper.

## D  RELATIONSHIPS TAXONOMY

Figure 9 presents the type of relationships considered in the analysis of ontological relationships between the candidates and a given target phenotype (Orange). If the candidate is not in the same branch from the ontology root (*Phenotypic abnormality*), we consider that no path exist between them, as the first level of the ontology refers to very different classes of abnormalities (e.g.: Abnormality of the musculoskeletal system, Abnormality of the nervous system, Abnormality of the cardiovascular system, ...).

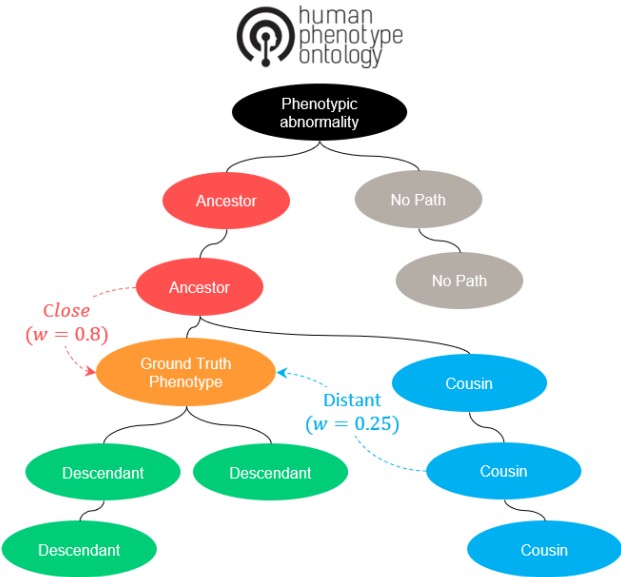

Figure 9: Relationships Taxonomy. Given a phenotype (orange), other nodes in the taxonomy are categorized as descendants (green), ancestors (red), cousins (blue), or outside the branch (grey). Relative closeness is computed using the relationship scores defined in Section 5.2.

While *Ancestor* and *Descendant* are common types, *Cousin* is a broader notion of relationships referring to candidates that have a common ancestor with the target. Finally, a *Close Relationships* type refers to candidates whose relationships score (Equation 4) with the target is above a threshold of 0.7.

## E  DATA VERIFICATION

As described in Section 9, training data were generated using ChatGPT-4o-mini to produce sentences from HPO phenotypes with a crafted prompt 17. The generated data were manually evaluated by our clinicians.

We first randomly sampled 50 sentences ensuring coverage across different branches of the HPO ontology. Each sentence was independently reviewed by two clinicians (blind annotation). The annotation guidelines were as follows:

- Meaningfulness: The sentence should be meaningful and coherent.
- Phenotype Reference: The sentence should refer to the input phenotype, either explicitly or implicitly.
- Clinical Realism: The sentence should resemble real-world clinical report formulations.

Annotators selected among three labels: 1 for sentences fully meeting all criteria, 0 if none were met, and 0.5 for partial or ambiguous phenotype references. To assess agreement on this ordinal scale, we used quadratic-weighted Cohen's kappa, yielding a score of 0.64, indicating strong reliability despite the small, imbalanced dataset. Notably, 88% of sentences were rated as high quality. The Cohen's Kappa of 0.64 suggests a strong reliability beyond chance, especially with such small annotation set and label imbalance. The confusion matrix (Figure 10) shows strong agreement on the highest score (1), and most disagreements occur between adjacent categories (e.g., 1 vs. 0.5), suggesting that when annotators differ, they still tend to assign similar quality levels.

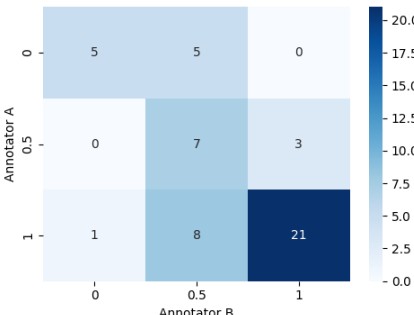

Figure 10: Annotation Confusion Matrix. The class distribution is imbalanced, with most sentences labeled as high-quality (class 1). Despite this, annotators show strong agreement across classes.

We acknowledge that 50 sentences represent a small sample, but the randomized selection across diverse ontology branches and the high level of agreement observed provide a reasonably robust basis for estimating overall data quality.

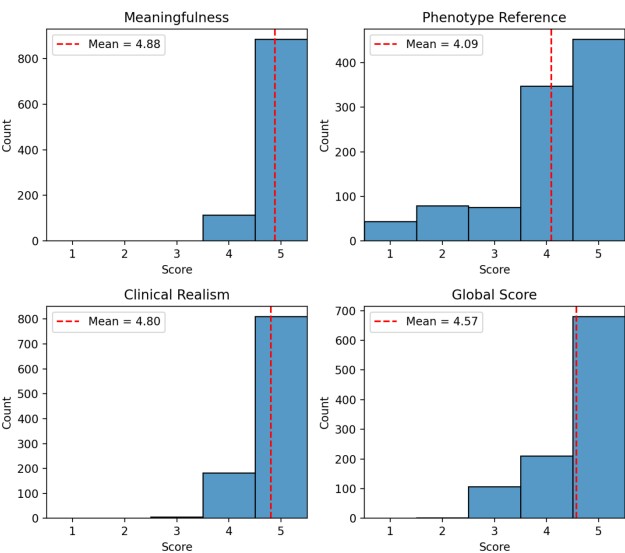

Figure 11: Distribution of LLM-as-a-judge scores across criteria. Scale are from 1 (poor quality) to 5 (excellent quality). Red lines represent the means.

To further support this assessment, we conducted an LLM-as-a-judge evaluation on a larger set of 1000 sentences across the three criteria (Figure 11). While the results should be interpreted with caution, they reinforce the observation that the vast majority of generated sentences are of high quality.

# F ADDITIONAL RESULTS

*Cross-Ontology Evaluation*

To assess the generalizability of HyperRAG beyond a single ontology, we conducted additional experiments using a hyperbolic model fine-tuned on the SNOMED ontology instead of HPO. This cross-ontology evaluation tests whether our approach can leverage hierarchical structure from different sources and whether combining ontologies can improve retrieval and ranking. While both ontologies are biomedical, this experiment provides an initial step toward demonstrating the robustness and adaptability of HyperRAG to diverse hierarchical knowledge bases. Future work will extend this analysis to non-biomedical domains.

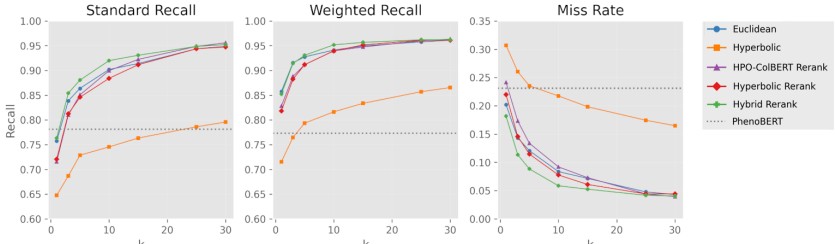

Figure 12: Recall and Miss Rate on the ID-68 dataset using a hyperbolic model trained on SNOMED ontology.

As shown in Figure 12, the SNOMED-based hyperbolic model underperforms compared to the HPO-based hyperbolic model, which is expected given that the target phenotypes are defined within the HPO ontology. Interestingly, the hybrid approach exhibits a slight improvement on the ID-68 dataset. This counterintuitive result suggests that, despite the SNOMED model being less domain-specific, it may introduce complementary information that is not captured by the HPO-based hyperbolic models. The combination of semantic similarity and the broader, more general structure encoded by SNOMED embeddings could help differentiate candidates in challenging cases, leading to improved ranking performance.

Conversely, the hybrid model demonstrates reduced accuracy with SNOMED on the CHU-50 dataset. Figure 13 shows recall and miss rate for CHU-50 dataset when using SNOMED hyperbolic model. As expected, the performance is lower than with the HPO hyperbolic model. However, we can note that the miss rate is better with the hybrid approach, suggesting that the hyperbolic model's structure (even when not domain-specialized) may still be effective at ensuring broad coverage of the ontology, due to the hierarchical nature of hyperbolic embeddings.

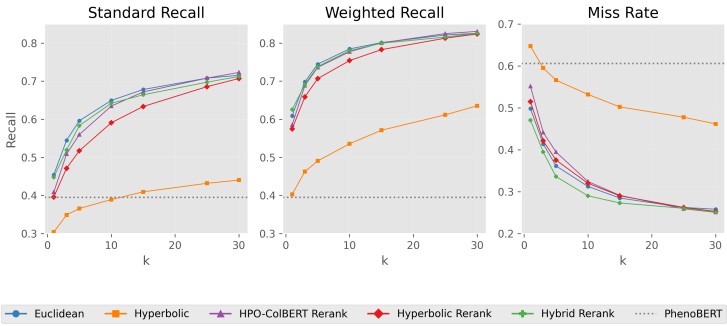

Figure 13: Recall and miss rate on the CHU-50 dataset using a hyperbolic model trained on SNOMED ontology. Standard (exact match) and weighted recall (considering HPO hierarchy) are shown for raw RAG outputs (Euclidean, Hyperbolic) and reranked results (HPO-ColBERT, Hyperbolic, Hybrid) across *Top-k* candidates. Miss rate indicates the proportion of ground truth phenotypes not retrieved.

These results highlight the potential for HyperRAG to integrate information from multiple ontologies, supporting its applicability to a wide range of hierarchical entity linking tasks.

*CHU-50 Ontology Metrics*

The average number of hops between candidates and target phenotypes is shown in Figure 14. The rank scale is up to 50 so the robustness of the hyperbolic model for higher ranks is highlighted.

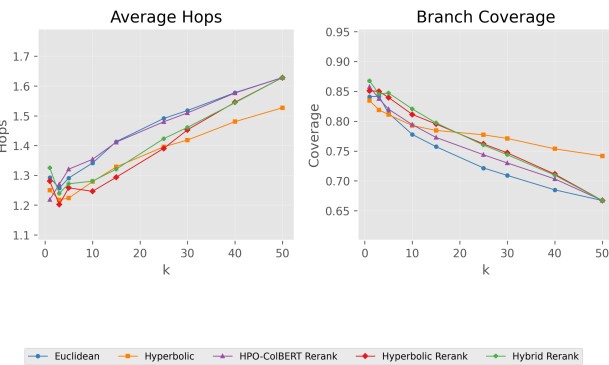

Figure 14: Ontological Structure Metrics on the CHU-50 dataset across *Top-k* candidates. Left plot represents the average number of hops between the candidates and the target phenotypes. Right plot shows the proportion of candidates within the ontology branch of the target phenotypes.

Finally, the distribution of relationships types for the CHU-50 dataset is shown in Figure 15.

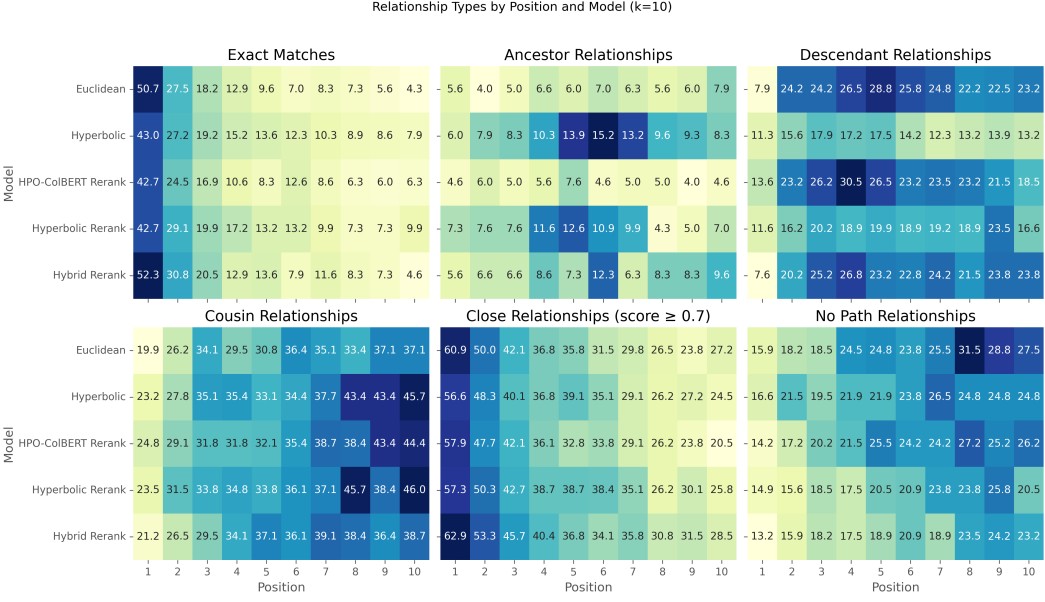

Figure 15: Distribution of relationship types by position and model for Top-10 candidates (k=10) on CHU-50 dataset. The heatmaps show percentages for six relationship categories (Appendix D) between predicted and ground truth phenotypes. For closeness relationships, only those with scores above 0.7 (Equations equation 4 and equation 5) are considered. Rows correspond to different models, and columns represent candidate positions ranked from 1 to 10. Higher values indicate stronger presence of the relationship type at the given rank and model.

## G PROMPTS

The prompt used for span detection step is shown in Fig. 16.

---

**Span Detection**

You are an experienced clinician with an exhaustive knowledge of human phenotypes ontology.
Given **{text}**, you must identify the spans related to possible phenotypes, either explicitly or implicitly.
You should keep in the span all words related to the phenotype that should be informative (such as negation or adjective).
You may reformulate the span if needed. If you don't detect any span or if you don't know, don't try to make up an answer, just write 'None'.

---

Figure 16: Full prompt used for span detection (variable in blue).

Late-interaction training data are built through automatic generation of sentences from HPO. This allows to have clinically-relevant sentences for all phenotype labels and synonyms from the ontology. In addition, we crafted prompt so that implicit and measurements references are generated. The corresponding prompt is shown in Fig. 17.

---

**HPO Sentence Generation**

For each HPO label, produce **{sentence_count}** purely observational sentences, referencing the phenotype explicitly or implicitly.
Requirements

- Use both the main HPO label and all provided synonyms for explicit references.
- At least **{implicit_count}** sentences must be implicit references (avoid using the label or synonyms).
- Ensure diversity in perspective or detail across sentences (as if from different medical domains and contexts).
- Ensure diversity in sentence openings.
- Use first-person ("I") or neutral ("we") style; do not include any titles (e.g., "Dr. Smith").
- At least two sentences must use passive voice.
- Vary sentence structure, wording and length (some short, some medium, some extended).
- Avoid overuse of "the patient" ($\leq$ 2 uses). Use pronouns ("he," "she," "they") or a fictitious name (first name or "Mr./Mrs." + last name) for diversity.
    - At least 1 sentence must use a fictitious name instead of "the patient" or a pronoun.
- Occasionally include measurements/tests (e.g., mg/dL, < 1st percentile), up to **{measures_count}** total.
- No interpretive language ("suggesting," "indicative of"); only factual observations.
- No professorial explanations; just present observations.
- Return a bulleted list ("- "). Clinical shorthand is fine.

Context
- **HPO label: {hpo_label}**
- **Definition: {definition}**
- **Synonyms: {synonyms_str}**

---

Figure 17: Full prompt used for automatic generation of sentences from HPO (variables are shown in blue).

As the goal was to generate spans related to phenotypes, we further query a LLM from the previously generated sentences to get the spans related to the corresponding HPO labels. The prompt used in this step is described in Fig. 18.

---

**Span Detection from Generated Sentences**

Extract the text span that best describes or indicates the phenotype "**{phenotype_data[' hpo_label' ]}**" from this sentence.
Requirements

1. Capture the complete clinical observation or symptom

2. Include relevant context that helps understand the phenotype

3. Be specific to the actual medical condition

4. Be concise while maintaining clinical accuracy

5. Exclude patient names, temporal markers, or examination context

6. Focus on the actual phenotypic finding

**Sentence:** "**{phenotype_data[' sentence' ]}**"
Extract only the relevant span without any additional commentary.

---

Figure 18: Full prompt used for span detection from generated sentences (variables are shown in blue). The object *phenotype_data* refers to a dataframe with the original HPO phenotype (*hpo_label*) and the corresponding generated sentence (*sentence*).

Finally, the prompt used for the LLM-as-a-Judge evaluation is presented in Fig. 19.

---

**LLM-as-a-Judge Evaluation**

You are an expert clinical language model evaluating sentences linked to HPO terms. Evaluate the sentence below using these criteria. Each score must be an integer from 1 to 5 (5 = excellent, 1 = poor):

1. **Meaningfulness** – Is the sentence meaningful and coherent?

2. **Phenotype Reference** – Does the sentence refer to the phenotype "**{hpo_label}**" (**{hpo_id}**), explicitly or implicitly?

3. **Clinical Realism** – Could this sentence appear in a real clinical report?

Then provide a **global score** (also 1–5) summarizing the overall quality of the sentence with respect to all three criteria.

Respond ONLY in this JSON format, filling in each `<score>` with your evaluation:

```
{
  "meaningfulness": <score>,
  "phenotype_reference": <score>,
  "clinical_realism": <score>,
  "global_score": <score>,
  "comment": "Brief explanation"
}
```

**Sentence:** "**{sentence}**"

---

Figure 19: Full prompt used for LLM-as-a-Judge evaluation.

