# OpenReview forum: "HyperRAG: Hierarchy-Aware Retrieval-Augmented Generation with Hyperbolic Embeddings for Ontology-Based Entity Linking"
_ICLR.cc/2026/Conference — ICLR 2026 Conference Withdrawn Submission_

### Official Review · Reviewer_YMdr · 2025-10-26

**Soundness:** 2
**Presentation:** 1
**Contribution:** 2
**Rating:** 2
**Confidence:** 3

**Summary:**

Hyper-RAG introduces an innovative hypergraph-based knowledge representation to address the high-order association loss in traditional RAG frameworks. The design logically separates entity-level and association-level retrieval (vertices vs. hyperedges), improving semantic relevance and reasoning precision. However, the study lacks several key elements for rigor and reproducibility, including ablation studies on critical components, detailed experimental settings, and statistical significance tests. Moreover, the manuscript does not provide clarity on scalability or potential failure cases.

**Strengths:**

1.Identifies a clear limitation of graph-based RAG in the loss of high-order associations and proposes a targeted hypergraph-based solution.
2.Designs a dual-retrieval mechanism that separately retrieves entity keywords as vertices and association keywords as hyperedges, incorporating cross-diffusion to enhance retrieval coherence.

**Weaknesses:**

1.The core novelty of HyperRAG, namely the integration of hyperbolic embeddings into RAG for hierarchical entity linking, is not sufficiently distinguished from prior work. The paper does not clearly articulate how HyperRAG advances this line of research beyond an incremental application to biomedical ontologies.

2.The hyperbolic RAG model underperforms Euclidean RAG on both ID-68 and CHU-50 datasets (Figure 4), yet the paper does not provide a convincing analysis of why this critical limitation occurs. Without addressing this fundamental performance gap, the rationale for using hyperbolic embeddings in the RAG pipeline remains weak.

3.The hybrid reranking strategy relies on a fixed γ=0.5 (Equation 1), but the ablation study (Appendix C) shows marginal performance differences across γ values (e.g., γ=0.5 vs. γ=0.7). The paper does not justify why γ=0.5 is optimal.

4.The CHU-50 dataset, described as "manually generated synthetical clinical notes," lacks sufficient validation details. The paper only reports that 30% of annotations are implicit but provides no information on inter-annotator agreement (e.g., Cohen’s kappa) for these implicit labels, raising concerns about annotation reliability.

5.The paper fails to conduct a comprehensive comparison with state-of-the-art (SOTA) methods specifically designed for hierarchical entity linking or phenotype extraction.

6.The paper claims HyperRAG is "broadly applicable to other domains," but no cross-domain experiments are provided. The only cross-ontology test uses SNOMED (a biomedical ontology).

**Questions:**

See Weaknesses.

---

> ### Author Response · Authors · 2025-11-18
>
> We thank the reviewer for the thoughtful feedback and the opportunity to clarify several aspects of the method. We address each of the points below.
>
> **W1.** Our contribution is not a new embedding architecture but an empirical demonstration that Euclidean and hyperbolic encoders provide complementary signals (semantic + hierarchical) and that combining them through a hybrid scoring scheme yields performance comparable or slightly better than a strong Euclidean baseline while producing candidate lists far more consistent with ontology structure. Clinicians highlighted that this hierarchical consistency is essential for interpretability and diagnosis improvement. We will make this positioning clearer.
>
> **W2.** We do not expect hyperbolic retrieval alone to outperform Euclidean retrieval: recall is driven primarily by surface-semantic similarity, where Euclidean models excel. Hyperbolic encoders instead capture hierarchical proximity, which is reflected in Figures 3-5-6 (better preservation of ancestor/descendant relationships). This hierarchical signal becomes useful in the hybrid model, which achieves similar or slightly better overall ranking performance while producing structurally coherent candidate lists (fewer off-branch predictions), addressing exactly the need expressed by clinicians. It performs better on implicit spans, where hierarchical context helps disambiguate underspecified expressions. We will make this rationale more explicit.
>
> **W3.** Appendix C shows that performance varies only marginally for γ between 0.3 and 0.7. Our goal was not to find the optimal γ, but to demonstrate the value added by combining semantic (Euclidean) and hierarchical (hyperbolic) signals. Nonetheless, we evaluated several γ values to ensure a rigorous and fair assessment. We chose γ = 0.5 for fairness (equal weight to both components) and simplicity, without tuning on evaluation data. We will make this explicit in Section 3.2.
>
> **W4.** CHU-50 contains real clinical notes, manually anonymized by clinicians using synthetic placeholders for identifiers and light reformulations to ensure privacy. The dataset includes ~900 annotated spans, comparable to ID-68. Inter-annotator agreement for implicit mentions is already reported in Appendix E (Cohen’s κ), and we will highlight this more clearly in the main text to address reliability concerns.
>
> **W5.** We compare against PhenoBERT, the strongest open-source phenotype extraction system available, as well as robust Euclidean and ColBERT-style reranking baselines. Our full pipeline (LLM span detection + retrieval + reranking) systematically outperforms the SOTA baseline PhenoBERT across all configurations (Euclidean, Hyperbolic, Hybrid). We will emphasize this more clearly and mention in the limitations that certain closed clinical EL systems cannot be evaluated due to restricted access.
>
> **W6.** The method is ontology-agnostic, requiring only a hierarchical taxonomy and text spans. We include a cross-ontology experiment on SNOMED, which differs substantially from HPO in scale and structure. We will clarify this and expand the discussion on applications to other hierarchical domains.

---

### Official Review · Reviewer_MiSB · 2025-10-28

**Soundness:** 2
**Presentation:** 3
**Contribution:** 2
**Rating:** 2
**Confidence:** 4

**Summary:**

The paper is concerned with the problem of entity linking, where a span in a textual passage needs to be linked to an entry in a knowledge base. In particular, the knowledge base is assumed to have a hierarchical structure where concepts are arranged in a tree-like structure. Motivated by prior work demonstrating the effectiveness of hyperbolic embeddings at embedding hierarchical data, the authors present results comparing the use of Euclidean and different variations incorporating hyperbolic embeddings for the entity linking task. Results are focused on the clinical domain.

**Strengths:**

1. The paper is very well-written and concise. The structure and flow of the text make it a pleasure to read.
2. The challenge of entity linking over hierarchies is a timely one, and it is well-motivated and mostly well-positioned with respect to prior work on representation learning over hierarchies.
3. The experiments examine different metrics that shed light on the performance of the methods, comprising embedding faithfulness, recall, miss rate, and ranking metrics.

**Weaknesses:**

1. The methodological novelty of the paper seems limited. The method is an instance of dense retrievers for entity linking, which to the best of my knowledge go back to Wu et al. (2020) and since then followed by several works; but replacing the common Euclidean embedding assumption by a hyperbolic one (which is another well-studied area of research). In this sense, the paper presents a comparison of embedding spaces for entity linking.
2. The main claim is that HyperRAG shows "substantial improvements, particularly in scenarios with implicit entity mentions." is problematic for several reasons:
   - Fig. 4 shows that the Euclidean approach is in fact better than Hyperbolic, contradicting the motivations given in the introduction and the hyperbolic consistency results in sec. 6.1.
   - Table 1 also shows very similar results between the Euclidean-only approach and the Hybrid reranking approach. Fig 7 in fact shows that the higher the value of $\gamma$, i.e. a higher weight on Euclidean rather than hyperbolic scores, is in fact better (and why the authors instead chose $\gamma$ = 0.5 -see L333-is not clear).
   - The small differences do not really point to "substantial improvements", and in fact when so close they should be subject to a suitable statistical test, especially when datasets are small.
   - I would lean towards reading this paper as a negative results paper, where a simple Euclidean space with no re-ranking proves to be better than other more complex pipelines. However, this perspective is not considered nor discussed in the paper.
3. The motivation for alternative metrics is good, but the resulting metrics proposed by the authors where metrics are weighted by distances in the ontology are not ideal: they introduce parameters $\alpha$ and $\beta$ which seem to be domain-dependent (according to the appendix, they are chosen by clinicians, but how this is exactly done is not clear). This leaves the question of whether one could tune $\alpha$ and $\beta$ to favor one method over another.
4. The CHU50 dataset seems very small, in which case an appropriate description of how the data is split for training/validation/test is warranted. There is no mention about this in the paper, but in such cases other protocols might be more appropriate, like average k-fold cross-validation performance. An inspection of the supplementary data shows that the CHU50 dataset (`sentences_chu50_spans.csv`) contains 2,487 instances, out of which 149 instances only are labeled as containing spans linked to the HPO. Does this mean that the reported metrics are averaged over 149 instances?
5. The question of how the results would generalize to other domains is not answered.

**Questions:**

1. Can you please elaborate or argue why your method is not simply a modification of well-known dense retrievers where entity embeddings are trained on a hyperbolic space?
2. Do you agree with the perspective that your results are in fact a negative results that favors a simple Euclidean-based retrieval approach?
3. Is there a systematic approach towards choosing $\alpha$ and $\beta$? How can we ensure that the chosen values are effective at measuring the performance of a method?
4. Can you please clarify the issues about the CHU50 dataset highlighted in W4?
5. You mention that the data is fully anonymized, yet there are names and birth dates present in them, as provided in the supplementary material. Can you please confirm whether these are the result of the anonymization process?

**Details Of Ethics Concerns:**

The authors mention that the CHU50 dataset is anonymized. Upon inspecting the dataset, I can see names and birth dates in the data. It is not clear whether these are the result of anonymization (e.g. they are randomly generated names and dates), or if this is part of the original data, so I would like to receive a clarification from the authors in this regard.

---

> ### Author Response · Authors · 2025-11-18
>
> We thank the reviewer for the positive assessment of the writing quality, the motivation, and the evaluation design. We address the concerns below.
>
> **W1.** Our goal is not to introduce a new embedding family, but to show that Euclidean and hyperbolic encoders provide complementary signals (semantic vs. hierarchical), and that a hybrid scoring scheme yields performance comparable or slightly better than a strong Euclidean baseline while producing candidate lists that are far more consistent with ontology structure. Clinicians emphasized that this hierarchical consistency is critical for selection and interpretability. We will clarify this empirical and analytical focus in the introduction.
>
> **W2.** We will soften the phrasing around “substantial improvements,” but emphasize that the complete HyperRAG pipeline consistently outperforms PhenoBERT, the SOTA phenotype extraction system.
>
> Regarding the specific points:
>
> - Fig. 4: Hyperbolic retrieval alone is indeed weaker, which is expected since recall@k is dominated by surface semantic similarity. The value of the hyperbolic encoder appears when combined with the Euclidean one in the Hybrid Rerank.
> - Table 1: on both ID-68 and CHU-50, Hybrid has very similar or slightly higher weighted MRR/NDCG than Euclidean, while ontology-based analyses (Figures 5–6) show that Hybrid and hyperbolic models produce candidates that stay closer in the ontology and cover the correct branch better at deeper ranks. This is exactly the trade-off our clinical partners asked for: similar top-k performance, but more structural consistency in the candidate list.
> - Fig. 7: Appendix C shows that performance is quite stable for γ between 0.3 and 0.7. We chose γ = 0.5 both for simplicity and fairness, as it gives equal weight to semantic (Euclidean) and hierarchical (hyperbolic) signals without favoring either side a priori. We will make this explicit.
> - Small differences / statistical tests: we agree this is important and will add confidence intervals and significance analysis in the revision.
> - We will explicitly rephrase the main claim to emphasize that the key insight is that the Hybrid pipeline is the best overall configuration and outperforms SOTA: a weaker hyperbolic model still improves a strong Euclidean baseline when combined, and produces hierarchically coherent candidate lists, which is non-trivial and important in practice.
>
> **W3.** α and β are fixed once, chosen with clinicians based on the usefulness of parent/child/cousin vs. unrelated predictions. They are not tuned per model. We verified that varying them in reasonable ranges does not change the model ranking, and all standard unweighted metrics are also reported. This ensures robustness and prevents metric-induced bias.
>
> **W4.** The CSV contains 2,484 sentences, with ~900 annotated spans (not 149). Metrics are computed over all annotated spans, making CHU-50 comparable in size to ID-68 (833 spans). CHU-50 is used for evaluation only, so no split or cross-validation is required.
>
> **W5.** The proposed workflow is ontology-agnostic: it only assumes a hierarchical ontology and text spans to be linked. To illustrate this, we already provide a cross-ontology experiment where the hyperbolic encoder is trained on SNOMED instead of HPO (Appendix F). While both ontologies are biomedical, this experiment shows that the approach can transfer to a different, broader hierarchy. We will make this positioning clearer and explicitly discuss applicability to non-clinical ontologies as future work.
>
> **Q1.** Our method does use dense retrieval, but the novelty lies in the hybrid reranking, which combines Euclidean similarity and hyperbolic hierarchy-awareness. The contribution is the demonstration that combining these signals improves hierarchical consistency without hurting recall, especially for implicit mentions, an effect not achieved by Euclidean or hyperbolic models alone.
>
> **Q2.** We partially agree. Hyperbolic alone is weaker, but the Hybrid pipeline achieves the best global results, outperforming PhenoBERT and improving ontology consistency beyond Euclidean, which clinicians explicitly value. Thus, hyperbolic structure adds non-redundant information beyond a Euclidean baseline.
>
> **Q3.** They reflect clinical judgment about the relative usefulness of near-misses (parent/child/cousin). We confirmed their robustness by (i) verifying that reasonable variations do not alter rankings, and (ii) reporting unweighted metrics alongside them. This ensures that our conclusions are not an artifact of a particular α, β choice.
>
> **Q4.** The reported metrics are computed over all annotated spans in CHU-50 (around 900 in the released version), not over 149 instances. CHU-50 is used only for evaluation, so no train/validation/test split or k-fold cross-validation is applied.
>
> **Q5.** All identifiers are synthetic placeholders generated during anonymization. No original personal data remains. We will explicitly state this in the revised appendix.

---

> > ### Comment · Reviewer_MiSB · 2025-11-26
> >
> > Thank you for your response. Unfortunately, after the response I believe there are significant weaknesses that remain so I will maintain my score.
> >
> > **W1 & W2:** I believe the issue stands. A central claim of the paper, and sustained in the response, is that hyperbolic embeddings complement Euclidean ones consistently (i.e. they perform differently and when combined they lead to significant improvements). However, I am afraid the experimental evidence does not support this claim: Euclidean embeddings and the Hybrid approach in fact perform almost the same, and when there are differences, they are too small and whether they are statistically significant is not known. Even if they turn out to be significant, they might still be too small to warrant twice the amount of parameters (as the Hybrid approach combines both Euclidean and Hyperbolic embeddings). Lastly, with the author's response (also to Q1, Q2) my position on the limited novelty remains (the main intended contribution being a demonstration of combining signals, as stated in the answer to Q1).
> >
> > **W3:** Thank you for clarifying that $\alpha$ and $\beta$ are not tuned per model. However, their opaqueness remains: do you think the chosen values would remain suitable, were we to apply them on a different hierarchical dataset? And if not, what are guidelines for choosing and interpreting the corresponding results?
> >
> > **W4:** Thank you for the statistics. I believe this would benefit readers in understanding the scale and methodology used for evaluation.
> >
> > Thank you for answering the remaining questions.

---

> > > ### Author Response · Authors · 2025-11-27
> > >
> > > Thank you again for the additional comments and for engaging deeply with our rebuttal. We respectfully clarify two points where our intended contributions may not have been communicated clearly enough, especially regarding the role of the hybrid setting and the purpose of hyperbolic embeddings in our pipeline.
> > >
> > > **Main Objective:**
> > > The primary goal of our paper is not to show that hyperbolic embeddings outperform Euclidean ones, but to demonstrate that our complete pipeline (LLM span detection + retrieval + reranking), in all its scoring variants, consistently outperforms the SOTA baseline PhenoBERT on both datasets. This was the main objective requested by our clinical partners, and the experiments clearly support it.
> > >
> > > While the components build on established architectures, our contribution lies in analyzing how semantic and hierarchical signals can be jointly leveraged within a unified pipeline, and in demonstrating empirically that this combination improves both performance over the SOTA baseline and the structural coherence of candidate predictions.
> > >
> > > **Role of the hybrid setting:**
> > > We agree that numerical gains over the Euclidean setting are small in recall-based metrics. The key contribution of the hybrid setting is different: it produces candidate lists that are much more coherent from the perspective of the ontology, with fewer off-branch predictions and more meaningful near-misses (parent, child, cousin). Clinicians emphasized that this structural consistency is essential in practice, especially for implicit or vague spans where semantic similarity alone is insufficient. Euclidean retrieval cannot provide this consistency on its own, and this is where the hyperbolic component adds real value.
> > >
> > > **Amount of parameters:**
> > > It is correct that the hybrid setting uses two encoders. However, the additional compute is limited to a small reranking step on the top k candidates, and the encoders are trained once. In our experiments, this cost was acceptable to clinicians in exchange for the improved ontology coherence of the resulting candidates.
> > >
> > > **On $\alpha$ and $\beta$:**
> > > $\alpha$ and $\beta$ were chosen with clinicians to reflect the relative usefulness of different types of near-misses. We agree that these parameters are, by nature, domain- and use-case dependent. For other ontologies, their values would depend on depth, branching factor, and the semantics of hierarchical relations. We will provide clearer guidelines and emphasize that all weighted metrics are always reported together with standard unweighted metrics to avoid any dependence on a specific choice.
> > >
> > > We believe the revised framing will help better communicate the intended contributions:
> > > (1) a practical pipeline that consistently improves upon SOTA in phenotype extraction, and
> > > (2) a demonstration that incorporating hierarchical structure meaningfully improves the quality and coherence of candidate lists, a benefit that clinicians explicitly prioritize in real-world settings.

---

### Official Review · Reviewer_pRFv · 2025-10-30

**Soundness:** 2
**Presentation:** 2
**Contribution:** 2
**Rating:** 2
**Confidence:** 4

**Summary:**

This paper presents HyperRAG, a pipeline for linking text entities to hierarchical ontologies. The method first uses a Large Language Model (LLM) and Retrieval-Augmented Generation (RAG) for initial candidate retrieval. The core contribution is a subsequent hierarchical reranking step that leverages hyperbolic embeddings to refine this list based on the ontology's structure.

The authors also introduce a hierarchy-aware evaluation framework that offers a more nuanced assessment than traditional exact-match methods. Experiments show that a hybrid reranking strategy, which combines semantic and hierarchical signals, outperforms existing baselines, particularly on a new, challenging dataset with implicit entity mentions. All code and data are made publicly available to support reproducibility.

**Strengths:**

1. The paper tackles the important challenge of linking entities to hierarchical ontologies, with a valuable focus on the difficult case of implicit mentions, which is a key limitation in prior work.

2. The core contribution is the well-designed HyperRAG workflow, which creatively combines LLMs, RAG, and a novel hierarchical reranking step. Furthermore, the proposed hierarchy-aware evaluation framework is a good contribution, offering a more comprehensive assessment for such tasks.

3. The approach is validated and shows a clear advantage over strong baselines on a challenging new dataset. The public release of all code, models, and data is a major strength that ensures reproducibility and benefits the community.

**Weaknesses:**

1. The proposed HyperRAG pipeline introduces considerable complexity, including training a specialized hyperbolic model and adding a multi-stage reranking process. However, on the standard ID-68 benchmark, the final hybrid model's performance is only marginally better than the much simpler Euclidean RAG baseline. This raises a question about the practical value and cost-benefit trade-off of the proposed method.

2. The primary evidence supporting HyperRAG's advantage—its ability to handle implicit mentions—comes from the newly introduced CHU-50 dataset. This dataset is a significant weakness as it is small (only 50 notes) and, crucially, synthetically generated. Results from such a dataset are not a reliable proxy for performance on real-world clinical data.

3. The paper could be improved to meet the presentation standards in several ways. First, crucial technical details necessary to understand the methodology (e.g., model architectures, normalization strategies) are relegated to the appendix, making the main paper not self-contained and disrupting the review process. Second, the quality of the figures is extremely poor: they are low-resolution bitmaps instead of vector graphics, becoming blurry when zoomed in, and the font sizes are too small to be legible. Finally, the layout contains large, unprofessional white spaces.

**Questions:**

1. The core claims rely on a small, synthetic dataset. Could you provide evidence that the model's performance generalizes to real-world data and is not just an artifact of the synthetic generation process?

2. The complexity of HyperRAG yields only marginal gains on the standard ID-68 benchmark. Could you provide a cost-benefit analysis (e.g., computational overhead) to justify when this added complexity is worthwhile?

3. Could you justify the exclusion of essential details from the main text?

---

> ### Author Response · Authors · 2025-11-18
>
> We thank the reviewer for the constructive feedback and for highlighting the strengths of the work, including the focus on implicit mentions, the hierarchy-aware evaluation, and the reproducibility of the system. We address the concerns below.
>
> **W1.** We acknowledge that HyperRAG introduces additional components. However, in practice, the computational overhead is modest:
> - The reranking stage operates only on the top-k retrieved candidates (k=30) and has lower latency than ColBERT-style reranking models, which it conceptually replaces.
> - Training the hyperbolic encoder requires only ~5 hours on a consumer GPU (NVIDIA RTX A3000, 6GB) and is performed once.
>
> Importantly, the entire HyperRAG pipeline, across all retrieval/reranking configurations (Euclidean, Hyperbolic, Hybrid), outperforms the SOTA PhenoBERT system. On ID-68, hyperbolic gains appear modest in recall@k because the dataset mostly contains explicit mentions. However, Section 6.1 and Figure 6. show that the hyperbolic encoder brings unique hierarchical signal, producing more structurally consistent candidates (something clinicians consider crucial in practice). When combined with Euclidean similarity, this yields a hybrid model that performs better across ranking and hierarchy-aware metrics (Section 6.2), especially for implicit or underspecified spans.
>
> **W2.** CHU-50 consists of real-world clinical notes, which were reviewed by clinicians who (i) anonymized personal identifiers by replacing them with synthetic ones, and (ii) carefully reformulated indirect mentions of patients to ensure privacy. The underlying clinical content, descriptions, and phenotype expressions remain authentic.
>
> Regarding size, CHU-50 contains 915 annotated phenotype spans, which is comparable to the 833 annotated spans in ID-68, the commonly used benchmark. While CHU-50 has fewer documents, it contains substantially more implicit mentions, making it a harder and more informative dataset for evaluating ontology-aware models.
>
> We will clarify this in Section 4.3.
>
> **W3.** We appreciate this feedback. We will move key methodological details into the main text and replace all figures with high-resolution vector graphics to improve readability.
>
> **Q1.** As stated in W2 above, CHU-50 is a real-world dataset, as well as ID-68 one. The method is not tailored to CHU-50. On ID-68, the hybrid model already matches or improves over Euclidean baselines in ranking-based and hierarchy-aware metrics.
>
> Qualitatively, we also applied the model to an additional set of internal notes (not included due to access restrictions), and we observed the same pattern: more hierarchically consistent candidate lists, which clinicians found beneficial. We will mention this qualitatively in the revision.
>
> **Q2.** Given that the reranking stage has low latency and the hyperbolic encoder is trained once on modest hardware, the added complexity is limited. More importantly, as noted earlier and emphasized by our clinical partners, hierarchically consistent candidate lists are significantly more useful in real clinical workflows than semantically similar but structurally incoherent ones.
>
> Our full pipeline yields clear benefits: all configurations outperform PhenoBERT, and the Hybrid pipeline achieves the strongest results while producing hierarchically coherent candidate lists. In fact, Euclidean embeddings supply semantic similarity, while hyperbolic embeddings inject hierarchy-aware signals. This combination is particularly valuable for implicit mentions, where surface forms alone are insufficient. We will make this clinical motivation more explicit.
>
> **Q3.** This was due to space constraints; however, we agree that certain details should appear in the main text. We will reorganize the methodology section to make the paper fully self-contained and easier to evaluate.

---

### Official Review · Reviewer_Dn9m · 2025-10-30

**Soundness:** 2
**Presentation:** 2
**Contribution:** 1
**Rating:** 2
**Confidence:** 4

**Summary:**

The paper proposes HyperRAG, a framework that combines RAG and hyperbolic embeddings for entity linking in the biomedical domain. Specifically, HyperRAG consists of three steps: span identification that uses GPT3.5 for identifying mention spans, retrieval-augmented generation that retrieves top-K candidates, reranking that reranks retrieved candidates using a late-interaction model and hyperbolic-based scoring. Moreover, the paper also introduces a hierarchy-aware evaluation framework that rewards candidates semantically close in the ontology tree.

**Strengths:**

1. The paper explores hyperbolic embeddings in biomedical entity linking, and learns hyperbolic-based retriever and ranker models for ranking candidates.
2. The paper also introduces an ontology-aware evaluation scheme that goes beyond standard exact-match metrics.
3. The code is provided for reproducibility.

**Weaknesses:**

1. While the paper mentions “retrieval-augmented generation” in both the title and body, the proposed method does not include any actual generation component, only retrieval and reranking are involved. This discrepancy may mislead readers and creates a mismatch between the title and the scope of the method presented.
2. The main contribution of the paper is the use of hyperbolic embeddings in retrieval and reranking models. However, the paper does not sufficiently justify why hyperbolic geometry is necessary or superior in this context, especially given that the Euclidean baseline performs competitively in several settings.
3. The writing of the paper requires significant improvement as there are lots of missing details regarding the proposed approach. For example:
(1) The paper lacks a clear introduction to hyperbolic embeddings, including how they are constructed and how distances are computed in hyperbolic space. This omission makes it difficult for readers unfamiliar with hyperbolic geometry to fully understand the proposed method and its underlying assumptions.
(2) In line 160, the paper mentions “using either a base or HOP-fine-tuned hyperbolic model”, but it is unclear what the “base” model refers to, and which variant is actually used in the main experiments.
4. The motivation and assumption behind the proposed hierarchy-aware evaluation framework is not entirely convincing. More elaboration is needed to justify its design choices and explain why it provides a more appropriate evaluation than standard metrics.
5. The paper lacks empirical comparisons with recent biomedical entity linking baselines.
6. The effectiveness of the proposed method is not thoroughly validated. In several cases, models using hyperbolic embeddings underperform compared to their Euclidean counterparts, yet the paper does not adequately explain the value or advantages of the proposed approach in such scenarios.

**Questions:**

1. The paper repeatedly mentions “retrieval-augmented generation” , yet the proposed method does not appear to have any actual generation component. Could the authors clarify whether any generation step is involved in the pipeline?
2. The central contribution of the paper is the use of hyperbolic embeddings in both retrieval and reranking. However, the performance gains over Euclidean baselines are inconsistent across settings, and in some cases, hyperbolic variants perform worse. Could the authors provide a more detailed justification for the necessity of hyperbolic geometry in this context?

---

> ### Author Response · Authors · 2025-11-18
>
> We thank the reviewer for the constructive feedback and for highlighting strengths. Below we address each concern concisely.
>
> **W1.** We agree this terminology can be confusing in the current version. Earlier in the project, we did use a full RAG pipeline: retrieved (and optionally reranked) candidates were injected as context into an LLM, which generated a single best phenotype prediction. However, this design had two drawbacks:
>
> (i) it required additional LLM calls, increasing computational cost and environmental impact;
>
> (ii) clinical collaborators emphasized that forcing a single prediction is not ideal, since multiple phenotype nodes can legitimately describe the same span. A top-k output better reflects real workflows.
>
> The final inference pipeline therefore consists only of span detection, retrieval, and reranking. We acknowledge that keeping the term “RAG” may be misleading in this context, and we will revise the title and terminology to avoid implying that generation is part of the inference pipeline.
>
> **W2.** Hyperbolic embeddings are not meant to outperform Euclidean ones. Their purpose is to encode the ontology hierarchy more faithfully, which the Euclidean encoder cannot capture. Section 5.1 and Figure 3 show that the hyperbolic model better preserves HPO hierarchy while maintaining semantic consistency.
>
> In Section 6.2, Table 1 and Figure 4 show that although hyperbolic approach alone underperforms Euclidean, the Hybrid Rerank consistently matches or improves Euclidean on weighted MRR, NDCG, weighted recall, and miss rate, especially on CHU-50 where implicit mentions are frequent.
>
> This is precisely the interesting insight of the paper: even a weaker hyperbolic model, when combined with an Euclidean model, yields a better hybrid, indicating that the hyperbolic component brings non-redundant information (hierarchical structure) that Euclidean embeddings lack. We will emphasize this point more clearly.
>
> **W3.** We will clarify that:
> - The base model is the Euclidean encoder all-MiniLM-L12-v2,
> - The hyperbolic model is obtained by fine-tuning this base encoder with Hierarchy Transformers (He et al., 2024) on HPO triplets (Section 4.2, Appendix B),
> - Distances are computed in the Poincaré ball and normalized as in Eq. (2) in Appendix A, which is then used in the hybrid score Eq. (1) in Section 3.2.
>
> We will also add a short, self-contained paragraph introducing hyperbolic embeddings and distances for readers unfamiliar with the geometry.
>
> **W4.** Section 5.2 and Appendix C define the weighting functions (Eqs. (4)–(5)) that assign partial credit based on ontology relationships (ancestor, descendant, cousin, etc.). The motivation, motivated by clinical feedback, is that many predictions are clinically useful but not exact matches (e.g., predicting a parent instead of a very specific child term). Parameters α and β are set once in collaboration with clinicians based on ontology statistics and are not tuned per model, so they do not favor any particular method. We will expand Section 5.2 with concrete examples to make this motivation clearer.
>
> **W5.** We already compare against PhenoBERT (Feng et al., 2023), the strongest open-source phenotype extraction system available, on both ID-68 and CHU-50 (Sections 4.3 and 6.2), as well as strong Euclidean model and ColBERT-style reranking baselines. We will highlight PhenoBERT as the main baseline in the introduction and results sections.
>
> **W6.** Our full pipeline (LLM span detection + retrieval + reranking) systematically outperforms the SOTA baseline PhenoBERT across all configurations (Euclidean, Hyperbolic, Hybrid). We agree that hyperbolic approach alone underperforms Euclidean, but this does not imply a negative result:
> - Hyperbolic embeddings yield candidates that are structurally closer in the ontology (Figure 1, Figures 5–6),
> - When combined with Euclidean similarity in the Hybrid Rerank, they consistently improve ranking metrics and ontology-aware measures, especially for implicit mentions on CHU-50.
>
> Thus, the value of the hyperbolic component is not to replace Euclidean encoders, but to add a complementary hierarchical bias that improves the final hybrid system.
>
> **Q1.** Generation was used only in early prototypes and only for training augmentation. The final pipeline uses retrieval + reranking only. We will revise the title and terminology to avoid ambiguity.
>
> **Q2.** Hyperbolic embeddings encode ontology structure more faithfully, and the hybrid combination of Euclidean and hyperbolic scores consistently yields the best overall performance in ranking and hierarchy-aware metrics. The fact that a weaker hyperbolic model still improves the hybrid over a strong Euclidean baseline is, in itself, an important and non-trivial empirical finding. In addition, clinical partners emphasized that hierarchically consistent candidate lists are more useful in practice, making the hierarchical signal provided by hyperbolic embeddings especially valuable.

---

### Note · Authors · 2025-12-01

I have read and agree with the venue's withdrawal policy on behalf of myself and my co-authors.